# Traumatic brain injury reprograms lipid droplet metabolism shaped by aging and diet in Drosophila brain

**Stacey A. Rimkus**[1], **Barry Ganetzky**[2], **David A. Wassarman** [1]*

1 Department of Medical Genetics, School of Medicine and Public Health, University of Wisconsin-Madison, Madison, Wisconsin, 2 Department of Genetics, College of Agricultural and Life Sciences, University of Wisconsin-Madison, Madison, Wisconsin

* dawassarman@wisc.edu

## Abstract

Traumatic brain injury (TBI) initiates secondary cellular damage such as mitochondrial dysfunction, oxidative stress, and neuroinflammation. In neurodegenerative disorders, these stressors are associated with accumulation of lipid droplets (LDs) – organelles that store neutral lipids to provide energy and protect cells from lipid toxicity. However, the regulation of LD metabolism following TBI remains poorly understood. Using a *Drosophila melanogaster* model, we investigated how TBI influences LD accumulation, particularly in relation to aging and diet, other LD modulatory factors. Confocal microscopy of fly brains at one day after injury showed increases in both LD size and number. The rise in LD number occurred only in flies fed a carbohydrate-rich diet and was absent in those given a ketogenic diet (KD) or water, suggesting that glucose availability is necessary for LD formation post-injury and potentially underlying why KD and water do not elicit the deleterious outcomes observed with carbohydrates. Lipidomic analysis of fly heads further revealed elevated levels of triacylglycerol (TG) species typically stored in LDs, indicating enhanced lipid synthesis post-injury. By seven days post-injury, LD size and number returned to baseline levels observed in uninjured flies and remained stable through 14 days post-injury. However, by 21 days post-injury, uninjured flies showed a marked increase in LD number that was not observed in injured flies, although LD size increased in both groups. These findings suggest that TBI selectively impairs age-dependent production of new LDs without affecting the growth of existing LDs. Importantly, TG levels remained elevated in heads of injured flies, indicating that the reduction in LD number was not due to limited lipid availability. Together, our findings indicate that TBI acutely induces LD formation as a protective response but chronically impairs LD biogenesis, disrupting lipid homeostasis in an age- and diet-dependent manner that may contribute to neurodegeneration.

**Data availability statement:** All relevant data are within the paper and its Supporting Information files.

**Funding:** Research reported in this publication was supported by the National Institute of Neurological Disorders and Stroke of the National Institutes of Health under Award Number RF1NS114359 and by the UW-Madison School of Medicine and Public Health, Graduate School, and Department of Medical Genetics.

**Competing interests:** The authors have declared that no competing interests exist.

## Introduction

Traumatic brain injury (TBI), resulting from external forces applied to the brain, affects millions of individuals worldwide each year, representing a major public health concern [1,2]. TBI begins with a primary injury, followed by secondary injures that unfold over minutes to years [3–5]. Secondary injuries involve complex cellular and molecular processes that can lead to cognitive, emotional, and physical impairments as well as increase the risk of developing neurodegenerative disorders such as Alzheimer's disease (AD) and chronic traumatic encephalopathy (CTE) [6–8]. Despite extensive research, the pathophysiological mechanisms underlying both short- and long-term consequences of TBI remain largely unresolved.

Identifying the cellular and molecular events that occur within minutes to hours after a primary injury is crucial for guiding treatment strategies, as these early responses are believed to influence both short- and long-term outcomes. Among the most important of these early responses are mitochondrial dysfunction [9–12], oxidative stress [13–15], and neuroinflammation [16–18], which initially engages the innate immune system. Notably, these same responses are also implicated in neurodegenerative disorders, suggesting shared mechanisms and offering insights into TBI pathology [19–27].

One downstream consequence of mitochondrial dysfunction, oxidative stress, and neuroinflammation is the formation of lipid droplets (LDs), organelles that compartmentalize cytosolic lipids [28–40]. LDs serve two main functions: they store lipids as a source of metabolic energy that can be accessed when needed, and they protect cells by sequestering excess or damaged lipids that could otherwise be toxic [31]. Thus, LD accumulation may indicate increased lipid stress, including the presence of excess or toxic lipids. LDs consist of a core of neutral lipids, primarily triacylglycerol (TG) and sterol esters, surrounded by a phospholipid monolayer with associated peripheral and integral proteins that tend to be involved in lipid metabolism. Accumulation of LDs in the brain is influenced by a variety of biological and environmental factors, including aging and diet [32]. Aging may promote LD accumulation, potentially due to increased production of toxic lipids, while caloric restriction appears to limit this buildup by reducing lipid synthesis. Importantly, growing evidence suggests that LDs play a neuroprotective role in the face of mitochondrial dysfunction, oxidative stress, and neuroinflammation, acting as a cellular defense mechanism in response to metabolic stress [33–36].

LD formation in glia is neuroprotective in a *Drosophila melanogaster* model in which reactive oxygen species (ROS) are generated by selectively knocking down the *ND42* subunit of mitochondrial electron transport chain Complex I in photoreceptor neurons of the eye [37,38]. In this model, neuronal ROS induces production of toxic lipid peroxides that are transferred to glia where they are incorporated into LDs. This lipid transfer depends on apolipoproteins, which have been linked to both TBI and AD. Genetic studies show that the *apolipoprotein E4* (*apoE4*) allele is associated with worse outcomes in both conditions [39,40]. In addition to LD formation protecting neurons from damage caused by lipid peroxides, a study of

mammalian cells in culture indicates that astrocytes metabolize fatty acids in LDs through mitochondrial β-oxidation to generate energy in response to increased neuronal activity [41]. Direct evidence linking LD formation to TBI has emerged from several models. In a mouse TBI model, 4-hydroxynonenal (4-HNE); a byproduct of lipid peroxidation that impairs cellular function by forming covalent adducts with proteins, DNA, and phospholipids; accumulates in the hippocampus weeks and months post-injury, and LDs accumulate in hippocampal microglia – innate immune cells – months post-injury [42]. Early inhibition of mitochondrial fission reduces 4-HNE, LD formation, and neurodegeneration, implicating ROS production, a consequence of mitochondrial fission, in this response. Rodent TBI studies also consistently show that lipid peroxidation is an early and prominent feature of secondary injury cascades [43–45]. Finally, LD accumulation is observed in microglia in a zebrafish penetrating brain injury model and in postmortem cortical brain tissue from TBI patients [46]. Together, these findings support a neuroprotective role for LDs in TBI and highlight the need to understand how this mechanism intersects with other modulators of brain health that affect LD production, such as aging and diet.

To investigate effects of TBI on aging and diet mechanisms that control lipid and LD metabolism, we used a Drosophila TBI model in which brain injuries and associated polytrauma are induced using a High-Impact Trauma (HIT) device [47]. Our previous studies support a role for LD metabolism in the primary injury response. Genome-wide association studies (GWAS) revealed that single nucleotide polymorphisms (SNPs) in genes involved in LD metabolism are associated with early mortality following TBI, defined as death within 24 hours of injury [48,49]. Among these genes is *lipid storage droplet-1* (*lsd-1*), the Drosophila ortholog of human *Perilipin1* (*PLIN1*), which encodes an LD-associated protein involved in regulating lipolysis [50,51]. Additionally, neuroinflammation – triggered through innate immunity signaling pathways–is activated within 30 minutes of injury and remains elevated for at least 24 hours [52]. Inhibiting these pathways or specific downstream transcriptional targets, significantly reduces early mortality, implicating neuroinflammation as a major contributor to acute TBI-induced lethality [53,54]. Post-injury diet also plays a critical role in early mortality. Flies fed high-carbohydrate diets such as a standard cornmeal-molasses-yeast diet (CMYD) or ≥1 M glucose show significantly higher early mortality than those fed a high-fat, low-carbohydrate ketogenic diet (KD) or water, indicating that metabolism of carbohydrates – precursors for fatty acid and lipid synthesis – exacerbates early mortality [48,52,55]. Finally, flies that survive the initial 24 hours after TBI develop progressive neurodegeneration over the following weeks, underscoring the long-term effects of TBI in flies [47,49]. These findings suggest that investigating how TBI influences the effects of aging and diet on lipid and LD metabolism may reveal new strategies to improve TBI outcomes.

Using confocal microcopy, we observed that TBI triggered a rapid and substantial increase in both the size of individual LDs and the number of LDs per cell in fly brains. This effect was most pronounced in flies fed a high-carbohydrate diet after injury. Over the following weeks, the typical age-related accumulation of LDs was disrupted in injured flies. Furthermore, lipidomic analysis of fly heads shortly after injury showed an increase in TG species typically found in LDs. However, as flies aged, differences in the abundance of these lipid species no longer aligned with LD levels. These findings suggest that when exogenous carbohydrates are available, TBI initially causes an increase in lipids that are sequestered in LDs. However, over time, TBI impairs LD formation, possibly allowing age-related accumulation of lipids to contribute to long-term pathologies such as neurodegeneration.

## Results

### TBI acutely increases LD size and number

To investigate LD dynamics following TBI, we used confocal microscopy of dissected fly brains to compare the size and number of LDs between injured and uninjured flies. These experiments used a standard laboratory fly strain, *w^1118^*. Female flies were injured at 8 days old using a HIT device [47], and age-matched, uninjured female flies served as controls. Bodipy (boron-dipyrromethene), a fluorescent dye that has high specificity for neutral lipids, was used to stain LDs [56]; an

antibody to Repo (Reversed polarity) marked glia [57]; and the fluorescent DNA stain DAPI (4',6-diamidino-2-phenylindole) labeled nuclei, including those of glia and neurons (Fig 1A). For each brain, images were obtained in four regions within the central brain at a plane with the highest concentration of nuclei (Fig 1B).

At 1 day post-injury, the size and number of LDs appeared to increase substantially (Fig 2). Near identical results were obtained with Nile Red, a lipophilic fluorescent dye that partitions into neutral lipid compartments [58], thereby independently confirming the identity of LDs (S1 Fig.). Quantitation of at least 11 brains and 21,000 cells per brain for each condition showed that the area of individual LDs was larger in injured compared to uninjured flies (Fig 3A, Table 1A, S1 Data). Assuming that LDs are spherical, injury increased their average diameter by 20%, from 0.95 µm in uninjured flies to 1.14 µm in injured flies. Correspondingly, injury increased the average volume of LDs by 73%, from 0.45 µm$^3$ to 0.78 µm$^3$. Injury also significantly increased the number of LDs per cell as well as the total area of LDs per cell (*i.e.*, area of individual LDs X number of LDs per cell) (Figs 3B and 3C, Tables 1B and 1C, S1 Data). These data suggest that TBI impairs lipid breakdown and/or enhances lipid synthesis, resulting in larger LDs, and that it inhibits LD degradation and/or promotes LD formation, increasing their number. Additionally, the observed increases in size and number may involve altered fission and fusion dynamics of LDs [31]. Overall, these changes may reflect a coordinated metabolic response to cellular stress that promotes sequestering excess or peroxidized lipids, thereby reducing lipotoxicity.

## LDs interact with microtubules and ROS-producing mitochondria following TBI

LDs are dynamic organelles that interact with various cellular structures. In particular, LDs associate with microtubules to facilitate their directed movement and spatial distribution within the cytoplasm [59]. This microtubule-dependent trafficking is important for positioning LDs in proximity to organelles such as mitochondria. LD-mitochondria contacts are functionally significant, as mitochondria are a major source of ROS following TBI [5,60]. To determine the localization of LDs in relation to microtubules and mitochondria, we again used confocal microscopy. Microtubules were labeled using an antibody to α-tubulin, and mitochondria producing ROS were identified using a ubiquitously expressed

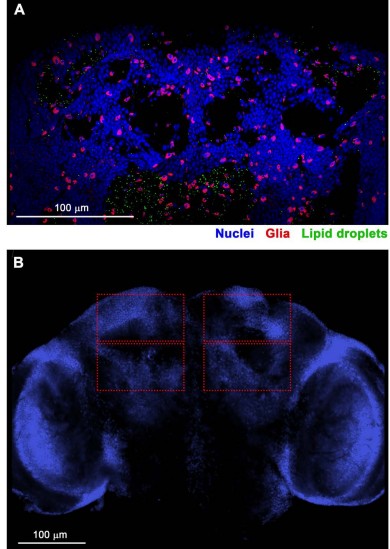

**Fig 1. Confocal microscopy of LDs in the central brain of flies. (A)** Posterior image of an uninjured 8-day old, female, *w$^{1118}$* fly brain at 60x magnification. **(B)** Anterior image of an uninjured 8-day old, female, *w$^{1118}$* fly brain at 20x magnification. Red boxes indicate regions of the central brain imaged at 100x magnification for quantitative LD analysis (Tables 1 and 2). DNA, DAPI (blue); glia, Repo (red); and LDs, Bodipy (green).

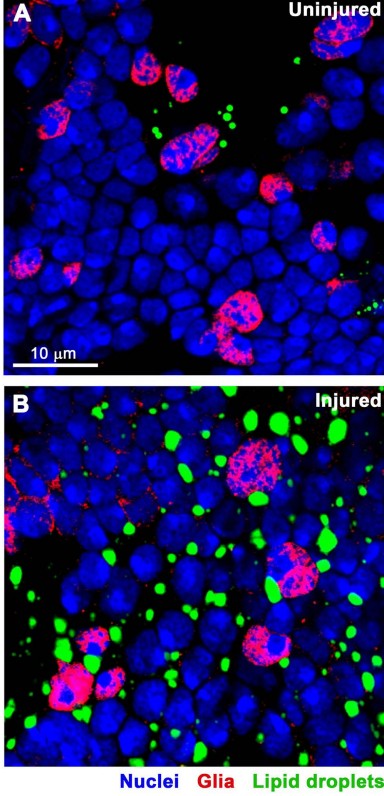

**Fig 2. TBI causes an acute increase in LD size and number in the brain.** Confocal microscopy images of the central brain in **(A)** uninjured and **(B)** injured 8-day old, female, $w^{1118}$ flies at 24 h after TBI. DNA, DAPI (blue); glia, Repo (red); and LDs, Bodipy (green). Fig 3 and "Tables 1A and B provide quantitation of LD size and number, respectively, based on images of this kind.

hydrogen peroxide ($H_2O_2$) sensor targeted to the mitochondrial matrix (tub-mito-roGFP2-Orp1) [61]. In both injured and uninjured flies, LDs were found to co-localize with microtubules in what appear to be axon bundles, supporting a role for microtubule-based trafficking in LD distribution (Figs 4A and 4B). Additionally, in injured flies, LDs were frequently positioned adjacent to ROS-producing mitochondria, suggesting that LD-mitochondrial interactions are functionally relevant in the oxidative environment post-TBI (Figs 4C and 4D).

## TBI acutely increases the abundance of TG species contained in LDs

The observed increase in LDs following TBI suggests that triacylglycerol (TG) metabolism is altered in response to injury. To investigate this, we performed high-throughput, semi-quantitative lipidomic analysis to assess changes in lipid species between injured and uninjured flies in an unbiased manner. Eight-day old $w^{1118}$ male flies were subjected to injury using the HIT device, fed CMYD, and analyzed 1 day post-injury. Age-matched, uninjured flies served as controls. For each condition, fly heads (n = 150) were collected in triplicate and analyzed using liquid chromatography-tandem mass spectrometry (LC/MS/MS) in both positive and negative ion modes. Lipidomic datasets acquired in each mode were normalized and combined to generate a comprehensive profile for downstream analysis (S2 Data).

To identify individual lipid species associated with TBI, we applied Partial Least Squares Discriminant Analysis (PLS-DA) to the lipidomic dataset, which consisted of 908 lipid species, including 258 TG species. This multivariate statistical approach allowed visualization of overall differences between injured and uninjured samples and the identification of

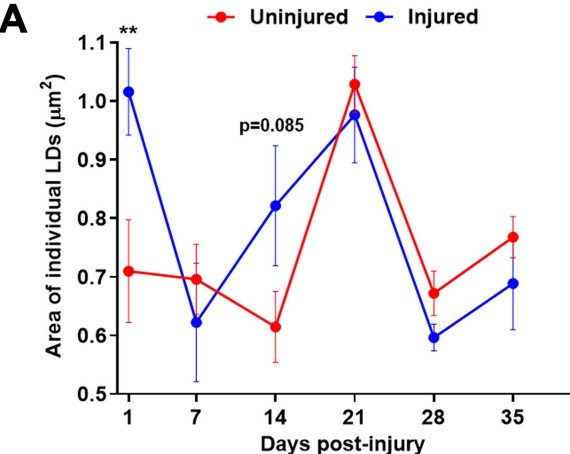

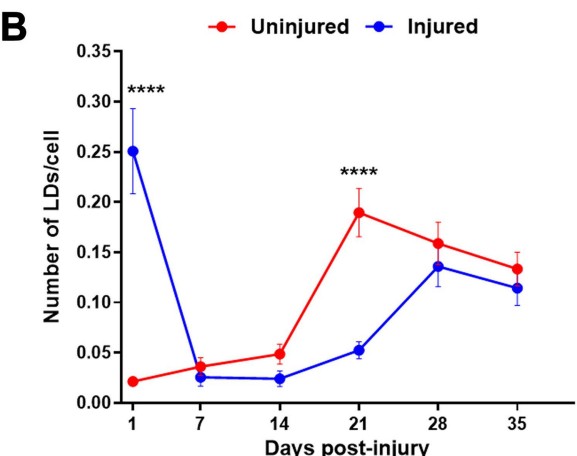

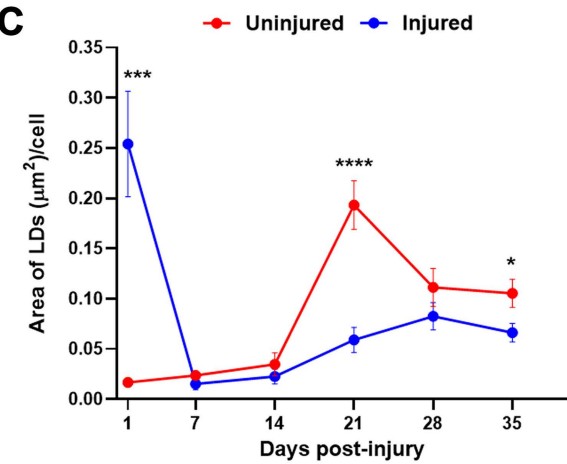

**Fig 3. TBI causes acute and chronic changes in LD size and number in the brain. (A)** Area of individual LDs, **(B)** number of LDs per cell, and **(C)** area of LDs per cell (*i.e.*, the data in **(A)** times the data in **(B)**) in uninjured (red) and injured (blue) $w^{1118}$ flies, as determined by analyses of confocal microscopy images. Each dot represents quantitation of >21,000 cells. Dots and error bars indicate the average and standard error of the mean (SEM), respectively, of data from ≥11 brains. In **(A)** and **(B)**, an unpaired Student's t-test was used to compare injured and uninjured flies at each of the days post-injury of 7-day old,

female, *w*<sup>1118</sup> flies, *p<0.05, **p<0.01, ***p<0.001, ****p<0.0001, and unlabeled time points (p>0.05). In **(C)**, an ordinary one-way ANOVA with Tukey's multiple comparison was used to compared the area of LDs per cell between injured and uninjured flies, *p<0.05, ***p<0.001, ****p<0.0001.

**Table 1. Effect of age on LD area and number.**

**A. Effect of age on individual LD area in injured and uninjured flies**

| Days post-injury | Area of individual LDs (µm²) | | p-value[a] |
| --- | --- | --- | --- |
| | Uninjured | Injured | |
| 1 | 0.71±0.09 | 1.02±0.07 | ** |
| 7 | 0.70±0.06 | 0.62±0.10 | ns |
| 14 | 0.61±0.06 | 0.82±0.10 | ns |
| 21 | 1.03±0.05 | 0.98±0.08 | ns |
| 28 | 0.67±0.04 | 0.60±0.02 | ns |
| 35 | 0.77±0.04 | 0.69±0.08 | ns |

**B. Effect of age on LD number per cell in injured and uninjured flies**

| Days post-injury | Number of LDs/cell | | p-value[a] |
| --- | --- | --- | --- |
| | Uninjured | Injured | |
| 1 | 0.02±0.00 | 0.25±0.04 | **** |
| 7 | 0.04±0.01 | 0.03±0.01 | ns |
| 14 | 0.05±0.01 | 0.02±0.01 | ns |
| 21 | 0.19±0.02 | 0.05±0.01 | **** |
| 28 | 0.16±0.02 | 0.14±0.02 | ns |
| 35 | 0.13±0.02 | 0.11±0.02 | ns |

**C. Effect of age on LD area per cell in injured and uninjured flies**

| Days post-injury | Area of LDs (µm²)/cell | | p-value[a] |
| --- | --- | --- | --- |
| | Uninjured | Injured | |
| 1 | 0.02±0.00 | 0.25±0.05 | *** |
| 7 | 0.02±0.00 | 0.02±0.01 | ns |
| 14 | 0.03±0.01 | 0.02±0.01 | ns |
| 21 | 0.19±0.02 | 0.06±0.01 | **** |
| 28 | 0.11±0.02 | 0.06±0.01 | ns |
| 35 | 0.11±0.01 | 0.07±0.01 | * |

[a]p-values are from comparisons between uninjured and injured flies at the indicated time point using an unpaired Student's t-test. ns=not significant, *p<0.05, **p<0.01, ***p<0.001, and ****p<0.0001.

lipid species that most strongly contributed to group separation. The resulting PLS-DA scores plot revealed clear clustering of samples based on injury status, indicating distinct lipidomic signatures between conditions (Fig 5A). The top 20 differentiating lipid species were all TGs (Fig 5B). The probability of this occurring by chance is exceedingly small (6.8 x 10⁻¹²), suggesting an important role for the TGs in the response to TBI. Furthermore, among these TGs, 97% of the fatty acid side chains had 12, 14, 16, or 18 carbon atoms, which are major energy substrates [62,63]. Seven of the lipid species were more abundant in injured than uninjured flies and the remaining 13 were more abundant in uninjured than injured flies, which may explain why the abundance of the TG class did not differ between injured and uninjured flies (S2A Fig., column 1). All seven TGs with increased levels post-injury contained at least two fatty acid side chains of 16:0, 16:1, 18:0, 18:1, or 18:2 (number of carbon atoms:number of double bonds) – the most abundant TG-associated fatty acids in LDs

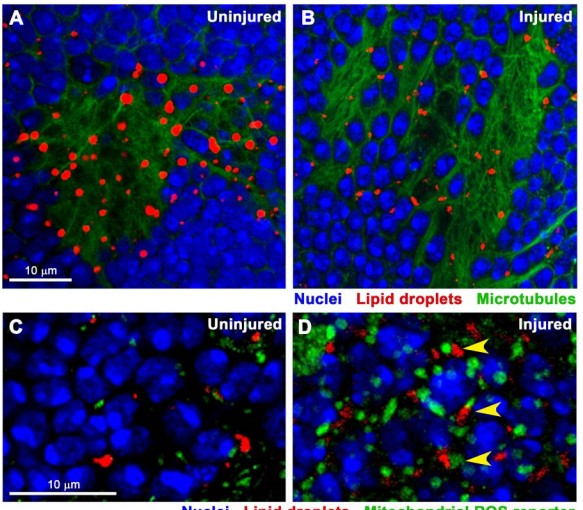

**Fig 4. LDs localize to microtubules and contact ROS-producing mitochondria in the brain after TBI.** **(A)** and **(B)** Confocal microscopy images of brains from injured and uninjured, female, *w1118* flies at 14 days after injury at 8 days old. DNA, DAPI (blue); glia, Bodipy (red), and microtubules, α-tubulin (green). **(C)** and **(D)** Images of brains from injured and uninjured, 8-day old, male, *w1118* flies carrying a transgene (tub-mito-roGFP2-Orp1) that expresses GFP in mitochondria that produce ROS [45]. Yellow arrows indicate LDs that are adjacent to ROS-producing mitochondria. DNA, DAPI (blue); mitochondrial ROS reporter, GFP (green); LDs, Nile Red (red).

[64,65] – and four of these TGs had three side chains from this group. In contrast, only four of the 13 TGs that decreased post-injury contained two of these side chains, and none contained three. These findings suggest that TBI acutely promotes the synthesis of TG species that are preferentially incorporated into LDs.

To assess global changes in lipid profiles following TBI, we compared the abundance of the 24 identified classes of lipid species between injured and uninjured flies (S2A Fig., column 1). These data indicate that injury caused the greatest fold increase in lysophospholipids (LPs) and fatty acid esters of hydroxy fatty acids (FAHFA). Acylcarnitines (CARs) had the greatest fold decrease in abundance post-TBI. Thus, TBI rapidly and broadly alters lipid metabolism.

## TBI impairs age-related LD accumulation

To investigate the long-term effects of TBI on LD dynamics, we repeated the microscopy assay at 7, 14, 21, and 35 days post-injury. Throughout this four week period, the area of individual LDs remained largely unchanged between injured and uninjured flies (Fig 3A, S1 Data). The dramatic decrease in LD size between 1 and 7 days in injured flies may result from TG breakdown through lipolysis or from LD fission. Lipolysis is more likely since the number of LDs per cell in injured flies also returned to the level of uninjured flies by 7 days post-injury (Fig 3B, S1 Data). Furthermore, while the substantial increase in LD size at 21 days in uninjured flies was also observed in injured flies (Fig 3A), the increase in LD number at 21 days was not (Fig 3B). However, by 28 and 35 days, the number of LDs in injured flies had risen to match that of uninjured flies (Fig 3B). Finally, total LD area per cell – reflecting both LD size and number – was substantially lower at 21 days post-injury and slightly lower at 35 days post-injury (Fig 3C, S1 Data). These findings suggest that over time, TBI selectively disrupts the formation of new LDs associated with aging, without altering the processes that regulate LD growth.

## TBI promotes accumulation of select TG and PE species during aging

We used lipidomic analysis to investigate the long-term effects of TBI on lipid dynamics. The experiment performed at 1 day after injury (Figs 5A and 5B) was repeated at 7, 14, and 21 days post-injury (Figs 5C and 5D). Across all samples,

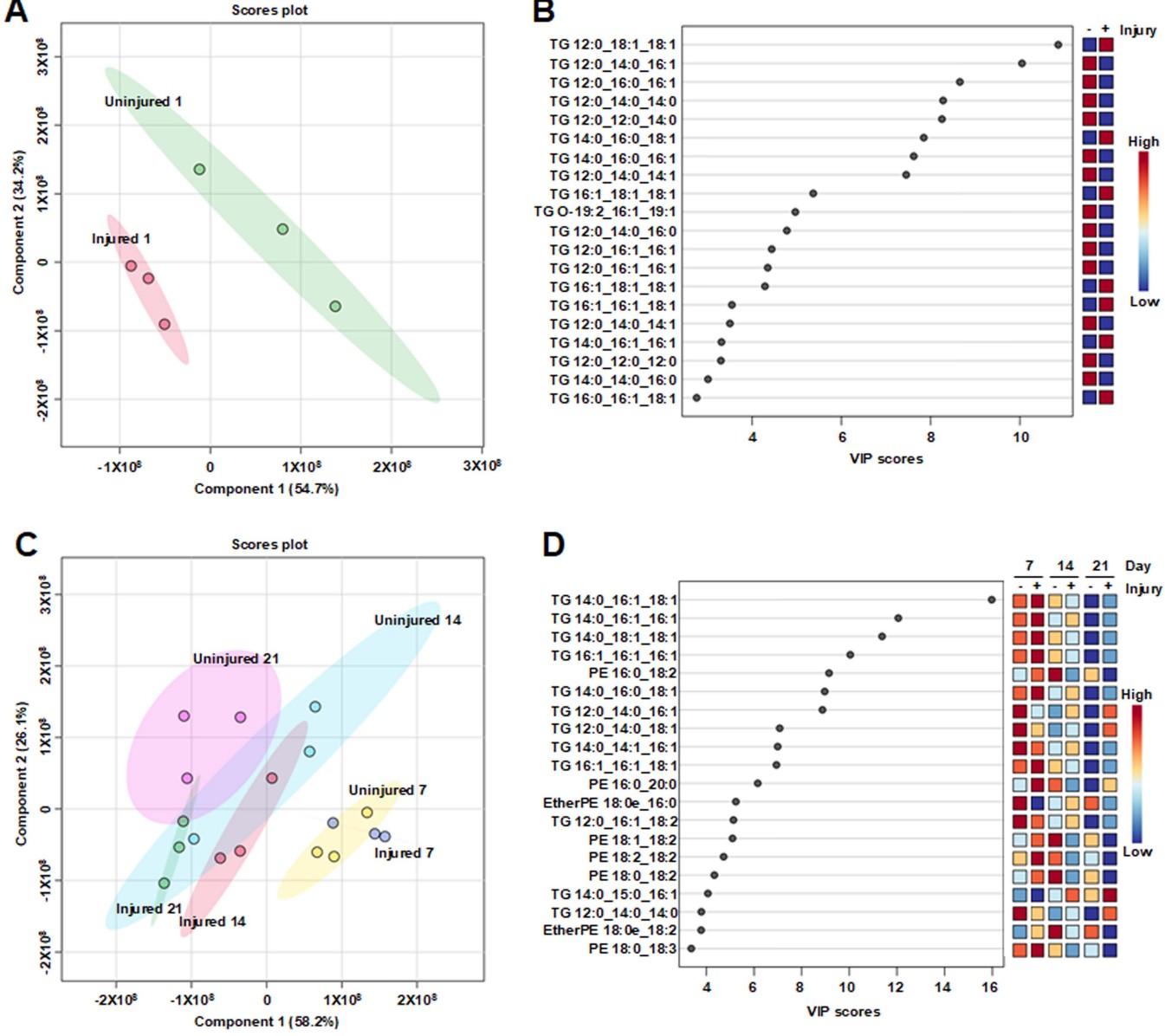

**Fig 5. Select lipid species in heads differentiate injured and uninjured flies.** Lipidomic analysis of heads was performed at 1, 7, 14, and 21 days post-injury of 7-day old, male, $w^{1118}$ flies to assess how injury alters the abundance of individual lipid species. PLS-DA analysis of injured and uninjured flies at **(A)** and **(B)** 1 day post-injury and **(C)** and **(D)** 7, 14, and 21 days post-injury. In scores plots **(A)** and **(C)**, small colored circles indicate independent samples, and colored ovals encompassing circles for a given condition indicate the 95% confidence limit. Components 1 and 2 percentages represent the percentage of the total variance in the data that is explained by the component. In **(B)** and **(D)**, Variable Importance in Projection (VIP), is a measure of how important each lipid species is in differentiating injured from uninjured flies. Heat maps show the relative abundance of each lipid species under the examined conditions: Red is high and blue is low abundance.

1401 lipid species were detected, including 365 TG species and 155 phosphatidylethanolamine (PE) species (S3 Data), some of which are components of LD membranes and contribute to membrane curvature and dynamics during LD formation and growth [66].

PLS-DA was used to identify lipids that differentiate injured and uninjured flies at 7, 14, and 21 days post-injury (Figs 5C and 5D). Component 1 of the scores plot organized the samples chronologically, suggesting that it reflects age-related lipid changes. Among the top 20 differentiating lipid species, 12 were TGs and 8 were PEs. The TGs were enriched in fatty acid side chains typically associated with LDs. In nearly all cases, these lipids decreased in abundance with age and at each time point were more abundant in injured than uninjured flies. This pattern contrasts with the age-related increase in LD abundance observed in uninjured flies (Fig 3C), suggesting that while TBI increases the abundance of lipid species typically stored in LDs as flies age, it also impairs LD formation, allowing these lipids to exert toxic effects.

Over time, the levels of select lipid classes, including cardiolipin (CL), lysophosphatidylinositol (LPI), phosphatidylinositol (PI), and phosphatidylserine (PS), steadily increased in injured compared to uninjured flies (S2A Fig., columns 2–4). Analyses of lipid class abundance at 7, 14, and 21 days post-injury relative to 1 day post-injury (S2B and S2C Figs., columns 2–4) indicate that the increase in CL, LPI, PI, and PS was driven by a decline in the level of theses lipids in uninjured flies, while levels in injured flies remained stable. These findings indicate that TBI has long-term effects on lipid metabolism in the head, which likely reflects changes in the brain because it occupies a substantial fraction of the head volume.

### TBI acutely activates LD accumulation mechanisms mediated by carbohydrate-rich diets

We previously found that the type of food consumed within 24 h after TBI influences early mortality, with significantly higher mortality for flies fed high-carbohydrate diets – CMYD or >1 M glucose – compared with those fed low-carbohydrate diets – ketogenic diet (KD) or water [48,55]. To investigate whether diet-related effects on early mortality might involve LD metabolism, we assessed LD size and number at 2 and 24 h following TBI of 8-day old, $w^{1118}$, injured and uninjured, female flies fed one of four diets post-injury: CMYD, 2 M glucose, a commercial KD, or water. The 2 h time point was selected because it is the earliest time at which effects of CMYD and water on early mortality have been observed [52].

As expected, diet affected early mortality after TBI: 24% for CMYD, 28% for glucose, 13% for KD, and 11% for water. However, early mortality among uninjured flies was < 1% across all diets, indicating that carbohydrate-rich diets do not directly cause early mortality but instead contribute to TBI-induced mechanisms that lead to early mortality. Accordingly, quantitation of at least 15 brains and 11,000 cells per brain for each condition, revealed that in both injured and uninjured flies, diet significantly affected individual LD size, LD number per cell, and total LD area per cell at 2 and 24 h (Fig 6 and Table 2). These data indicate that diet influences LD abundance through injury-independent and -dependent mechanisms. At 24 h, injury caused the most significant increase in LD area per cell in flies fed a carbohydrate-rich diet – CMYD or glucose (Table 2C) – consistent with the results shown in Figs 2 and 3 and Table 1C. The increase in total LD area per cell was largely driven by an increase in the number of LDs per cell (Table 2B). The underlying mechanism was activated by 2 h for glucose but not CMYD. The delay with CMYD may be due to the additional time required to digest complex carbohydrates to glucose. At 24 h, TBI had little or no effect on LD area per cell with KD and water diets, respectively (Table 2C), despite significant effects of these diets on LD size and number at 2 and 24 h (Tables 2A and 2B). These findings reveal a correlation between TBI-induced LD accumulation and early mortality across the different diets, suggesting that LDs either contribute directly to early mortality or arise as a result of processes that induce early mortality. However, it is important to note that absolute LD abundance per cell did not correlate with early mortality following TBI: flies fed KD or CMYD had similar LD area per cell at 24 h (Table 2C), yet experienced different levels of early mortality. This may reflect the dual functions of LDs: under CMYD, increased LD abundance may result from the sequestration of lipid peroxides, which are associated with early mortality; in contrast, under KD, the increase in LDs may reflect storage of undamaged lipids for β-oxidation, serving as an energy source that helps prevent early mortality [67,68].

## Discussion

Our findings shed light on how TBI alters lipid and LD metabolism in the context of aging and dietary stress – factors highly relevant to human TBI. Existing literature supports a three-step model of the brain's response to TBI. (Step 1:

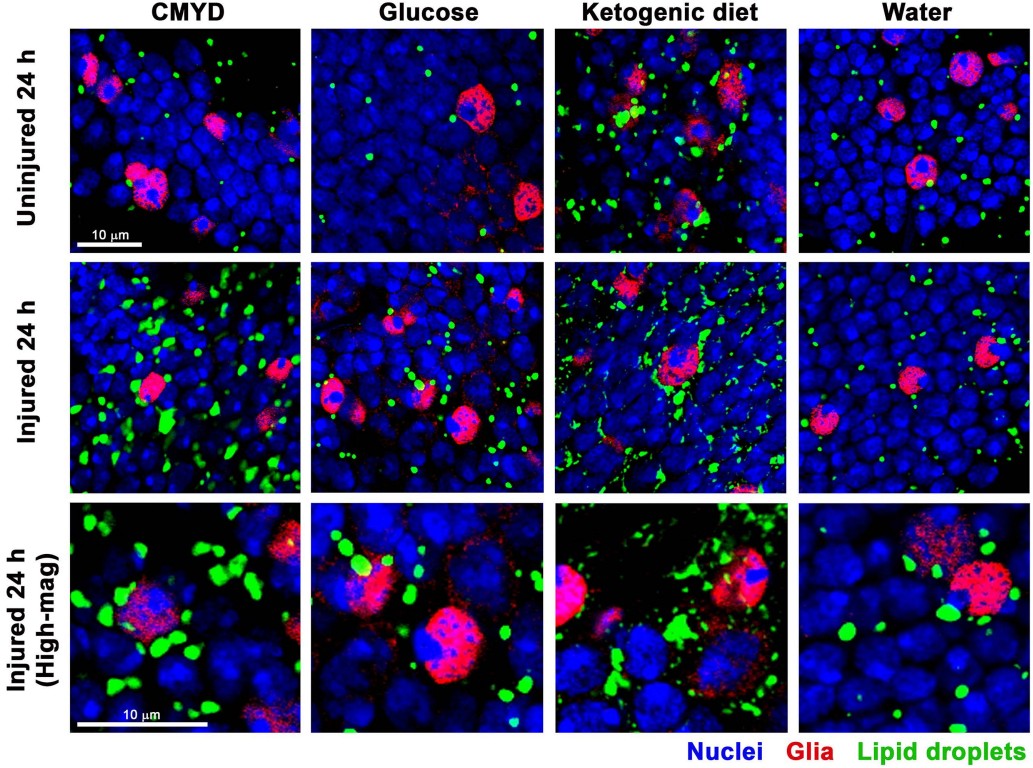

**Fig 6. Confocal microscopy of LDs in the central brain of flies under different diet conditions.** Injured and uninjured 8-day old, female, $w^{1118}$ flies (rows) were fed different diets (CMYD, 2 M glucose, ketogenic diet, or water (columns)) and imaged 24 h post-injury. Top row, uninjured flies (100x magnification), middle row injured flies (100x magnification), and bottom row, injured flies (200x digital magnification). DNA, DAPI (blue); glia, Repo (red); and LDs, Bodipy (green). "Tables 2A and B provide quantitation of LD size and number, respectively, based on images of this kind.

initial response) TBI acutely triggers mitochondrial dysfunction, oxidative stress, and neuroinflammation; (Step 2: lipid damage) this leads to rapid generation of ROS, lipid peroxidation, and production of toxic lipid species (*e.g.*, 4-HNE); and (Step 3: LD formation) cells respond by sequestering lipids into LDs to prevent lipotoxicity and by trafficking LDs along microtubules to mitochondria to generate energy (Fig 4) [37,38,41]. Our findings extend this model by showing that high-carbohydrate diets post-TBI exacerbate LD accumulation (Fig 6 and Table 2), likely due to providing excess substrates for lipid synthesis. In contrast, low-carbohydrate diets may mitigate production of toxic lipid species. Additionally, while age-dependent accumulation of LDs likely serves as a protective mechanism against cumulative oxidative and metabolic stress, our findings indicate that in an injured brain, lipids are initially sequestered into LDs (Figs 2 and 3 and Table 1), but over time, lipids accumulate to a greater extent (Fig 5) and are not properly sequestered into LDs (Fig 3 and Table 1), increasing the potential for lipotoxicity and contributing to neurodegeneration.

## Impaired trafficking of LDs along microtubules might contribute to CTE

The association of LDs with microtubules in axons of injured brains (Fig 4B) provides the first direct evidence, to our knowledge, that disrupted LD trafficking may contribute to the development of CTE pathology. CTE is characterized by microtubule disruption, impaired intracellular transport of cargoes, and abnormal hyperphosphorylation of tau, which destabilizes microtubules and compromises neuronal structural integrity [8]. Our prior studies implicated the microtubule-associated proteins Lissencephaly-1 (Lis-1) and Patronin in TBI outcomes [49]. Lis-1 regulates dynein, the microtubule

**Table 2. Effect of diet on LD area and number.**

**A. Effect of diet on individual LD area in injured and uninjured flies**

| | Area of individual LDs (µm²) | | | | | |
|---|---|---|---|---|---|---|
| Diet | Uninjured 2 h | Injured 2 h | p-value[a] | Uninjured 24 h | Injured 24 h | p-value[a] |
| CMYD | 0.61±0.03 | 0.60±0.03 | ns | 0.62±0.02 | 0.77±0.03 | *** |
| Glucose | 0.62±0.03 | 0.47±0.02 | **** | 0.72±0.04 | 0.67±0.03 | ns |
| Ketogenic diet | 0.67±0.02 | 0.64±0.02 | ns | 0.67±0.03 | 0.80±0.03 | ** |
| Water | 0.53±0.02 | 0.63±0.02 | ** | 0.61±0.02 | 0.73±0.04 | ** |
| p-value[b] | * | **** | | * | * | |

**B. Effect of diet on LD number per cell in injured and uninjured flies**

| | Number of LDs/cell | | | | | |
|---|---|---|---|---|---|---|
| Diet | Uninjured 2 h | Injured 2 h | p-value[a] | Uninjured 24 h | Injured 24 h | p-value[a] |
| CMYD | 0.08±0.01 | 0.11±0.02 | ns | 0.21±0.02 | 0.47±0.05 | **** |
| Glucose | 0.16±0.01 | 0.41±0.06 | *** | 0.09±0.01 | 0.34±0.03 | **** |
| Ketogenic diet | 0.27±0.03 | 0.49±0.06 | *** | 0.33±0.03 | 0.43±0.07 | ns |
| Water | 0.22±0.03 | 0.20±0.02 | ns | 0.25±0.03 | 0.20±0.02 | ns |
| p-value[b] | **** | **** | | **** | *** | |

**C. Effect of diet on LD area per cell in injured and uninjured flies**

| | Area of LDs (µm²)/cell | | | | | |
|---|---|---|---|---|---|---|
| Diet | Uninjured 2 h | Injured 2 h | p-value[a] | Uninjured 24 h | Injured 24 h | p-value[a] |
| CMYD | 0.05±0.00 | 0.07±0.01 | * | 0.13±0.02 | 0.36±0.04 | **** |
| Glucose | 0.10±0.01 | 0.19±0.03 | * | 0.07±0.01 | 0.23±0.02 | **** |
| Ketogenic diet | 0.19±0.02 | 0.32±0.04 | ** | 0.22±0.02 | 0.34±0.05 | * |
| Water | 0.12±0.01 | 0.13±0.02 | ns | 0.15±0.01 | 0.16±0.02 | ns |
| p-value[b] | **** | **** | | **** | *** | |

[a]Unpaired Student's t-test analysis of effects of an individual diet on LD properties in injured vs. uninjured flies at 2 or 24 h post-injury. ns = not significant, * p < 0.05, ** p < 0.01, *** p < 0.001, and **** p < 0.0001.

[b]Normal one-way ANOVA analysis of effects of diet on LD properties in injured and uninjured flies at 2 h and 24 h post-injury.

motor responsible for retrograde transport of many cargoes, including LDs [69], while Patronin stabilizes microtubules by protecting their minus ends from depolymerization [70]. SNPs in the *Lis-1* and *Patronin* genes are associated with early mortality following TBI, and mutations in *Lis-1* enhance early mortality and neurodegeneration following TBI [49]. Furthermore, reduced expression of *Lis*-1 is associated with a late stage of CTE [71]. Together, these findings suggest that microtubule-dependent regulation of LD localization – particularly their trafficking to mitochondria (Fig 4D) – is a key component of the cellular response to TBI and defects in the process may contribute to the development of CTE pathology.

## TBI rewires effects of aging on LD accumulation

We found that a primary brain injury induces dynamic changes in LD abundance that can manifest weeks after a primary injury. LD size and number increased significantly at 24 h post-injury but returned to baseline within 7 days (Fig 3 and Table 1). This resolution likely reflects recovery from the conditions that initially promoted LD accumulation – such as mitochondrial dysfunction, oxidative stress, and neuroinflammation – and may involve lipid breakdown by lipases or lipophagy, where LDs are engulfed by autophagosomes and degraded by lysosomes [72]. In the weeks following injury, LD number increased, consistent with prior observations in flies and mice [73,74]. However, the typical age-related increase in LD abundance was attenuated in injured flies compared to uninjured controls (Figs 3B and 3C). This difference was most pronounced at 21 days post-injured (*i.e.*, in 29 day old flies), when uninjured flies had significantly more LDs per cell (Fig.

3B) but similar LD size (Fig. 3A), suggesting that TBI impairs the formation of new LDs without affecting the growth of existing ones. This time point aligns with related physiological events in aging flies, suggesting that they may be functionally linked. First, brain lipid turnover peaks at around 25 days of age, as indicated by the ratio of newly synthesized lipids to total lipids during aging from 5 to 45 days [32]. Second, a transient spike in 4-HNE levels is observed in fly heads at approximately 38 days of age [75]. Third, flies injured at 0–7 day old and surviving the first 24 h exhibit normal survival for about 17 days, but begin dying at a higher rate than uninjured flies between 17 and 25 days of age [47,55]. Collectively, these findings suggest that TBI disrupts programmed changes in lipid and LD metabolism during aging, potentially contributing to neurodegeneration and other late-onset consequences of injury. In particular, a primary injury may acutely trigger LD metabolic events that typically occur later in life (*i.e.*, LD size and number in injured flies at 1 day post-injury was comparable to that of uninjured flies at 21 days post-injury) (Figs 3A and 3B and Tables 1A and 1B). This early onset may contribute to accelerated, age-associated neurodegeneration.

## TBI acts through diet-dependent pathways to influence LD accumulation

Our findings indicate that reducing carbohydrate intake – either by feeding KD or water immediately after TBI – may protect against injury-related damage by limiting production of toxic lipids, as reflected by the absence of new LD formation (Fig 6 and Table 2B). Similarly, carbohydrate-rich Western diets lead to worse TBI outcomes than KD in rodents [76]. Moreover, in a rat TBI model, KD-fed animals showed reduced oxidative stress and lower levels of 4-HNE-modified proteins at 6 and 24 h post-injury compared to those fed a high-carbohydrate, standard chow diet [77]. The protective effects of KD appear to be mediated by ROS scavenging and enhanced mitochondrial electron transport chain activity, suggesting that the absence of TBI-induced LD formation in KD-fed flies results from reduced upstream signals linked to mitochondrial dysfunction and oxidative stress.

Our prior data show that flies given only water post-injury experience substantially lower early mortality than those fed CMYD, with the protective effect most pronounced when water is provided between 2 and 8 h after injury [52]. Similarly, Davis *et al.* (2008) showed in a rat TBI model that 24 h of water-only fasting after injury compared to unrestricted access to a balanced diet reduced lesion volume, mitochondrial dysfunction, ROS production, and lipid peroxidation and improved cognitive function [78].

## Limitations of the study

A major limitation of this study is that it was conducted in a single genetic background. Our studies in flies, along with studies in rodents and humans, demonstrate that genetic background strongly influences the outcomes of TBI [48,49,52,79,80]. Rodents receiving identical primary injuries can exhibit markedly different outcomes depending on their genotype [81–84]. This variation underpins our GWAS analyses that identified associations between early mortality following TBI and SNPs in *lsd-1*, *Lis-1*, and *Patronin* [48]. In our experiments using 30 inbred fly lines from the Drosophila Genetic Reference Panel (DGRP) [85,86], early mortality following TBI varied substantially across genotypes [52]. Furthermore, factors such as age at the time of injury and post-injury diet influenced genetically distinct secondary injury pathways. For instance, while most DGRP fly lines showed higher early mortality when injured at an older age and when fed CMYD rather than water, DGRP439 was sensitive to age but not diet – indicating genetic-based resistance to the carbohydrate-driven early mortality pathway. In contrast, DGRP381 was diet-sensitive but not age-sensitive. These findings suggest that lipid and LD metabolism responses following TBI in $w^{1118}$ flies are unlikely to generalize across other genetic backgrounds. Understanding how genetics differences affect lipid and LD metabolism in flies may ultimately inform the development of personalized therapies for genetically diverse human populations.

Additionally, because flies that died were not examined, it remains unclear whether LD accumulation is protective or detrimental in the context of early mortality. The elevated LD levels observed in flies that survived one day post-injury is consistent with a protective role, potentially by sequestering toxic lipids and supplying energy to support neuronal

recovery. However, it is also possible that excessive LDs accumulation is detrimental, and that survivors had lower LD levels than those that died. To resolve this, it will be necessary to compare animals that will survive to those that are fated to die within one day following TBI. This is feasible in flies, as increased intestinal permeability after TBI reliably marks individuals that will die within one day [48,87]. Therefore, to determine the functional role of LDs, their levels should be measured in flies exhibiting intestinal barrier dysfunction to determine whether LD accumulation correlates positively or negatively with early mortality.

Finally, an important confounding factor in the interpretation of both the confocal microscopy and lipidomic data is that these analyses were performed on bulk tissue – whole brains in the case of microscopy and whole heads for lipidomic analysis. As a result, these data represent an average signal across multiple cell types, potentially obscuring cell type-specific responses to TBI. LDs are known to vary in size, abundance, and function across different brain cell types [37,41,88]. Following TBI, it is possible that LDs increase in size and/or number in some cell populations while decreasing in others or that certain lipid species are differentially regulated depending on the cell type. Such heterogeneity could not only lead to misinterpretation of the directionality or magnitude of the response but also mask meaningful biological effects. Our microscopy data (Figs 1, 2, 4, 6, and S1 Fig.) do not clarify whether TBI-, aging-, and diet-induced LD changes in LD size and number occur in neurons or glia. Demonstrating neuronal localization would be paradigm-shifting, as glia are currently considered the primary site of stress-induced LD accumulation [35,37,38,46,68,73,74]. To resolve these ambiguities, future studies incorporating cell type-specific labeling, sorting, or single-cell approaches will be critical to accurately characterize how TBI alters lipid and LD metabolism. Nonetheless, bulk tissue analyses presented here have been tremendously informative, providing foundational understanding of lipid and LD alterations after TBI and offering critical impetus for pursing more refined, cell type-specific investigations.

## Materials and methods

### Drosophila stocks and husbandry

Flies were maintained on a standard fly food, cornmeal-molasses-yeast diet (CMYD), at 25°C, unless otherwise stated. CMYD contained 30 g Difco granulated agar (Becton-Dickinson), 44 g YSC-1 yeast (Sigma), 328 g cornmeal (Lab Scientific), 400 ml unsulfured Grandma's molasses (Lab Scientific), 3.6 L water, 40 ml propionic acid (Sigma), and tegosept (8 g Methyl 4-hydroxybenzoate in 75 ml of 95% ethanol) (Sigma). $w^{1118}$ flies were maintained in our lab for many years, and $w^{1118}$; tub-mito-roGFP2-Orp1 flies [61] were obtained from the Bloomington Drosophila Stock Center (stock #67673).

### TBI with different post-injury diets

Flies were collected at 0–1 days post-eclosion, aged 7 days, and subjected in groups of 30 to TBI (4 strikes from a HIT device with 5 min between strikes), as described in Katzenberger *et al.*, 2013 [47]. Uninjured flies were manipulated the same as injured flies, but they were not subjected to strikes from HIT device. Injured and uninjured flies were transferred to vials with CMYD or a Whatman paper disc at the bottom that contained 200 µl of $H_2O$, 2 M glucose (Sigma), or a commercial ketogenic diet (KD) (Teklad ketogenic diet, TD.96355, Envigo) at 0.3 calories/200 µl. KD contained 173.3 g/kg casein, 2.6 g/kg DL-methionine, 586.4 g/kg vegetable shortening (Crisco), 86.2 g/kg corn oil, 88.0 g/kg cellulose, 13.0 g/kg vitamin mix (Teklad 40060), 2.5 g/kg choline bitartrate, 0.1 g/kg tertiary butylhydroquinone (TBHQ), 20.0 g/kg mineral mix (calcium phosphate deficient), 19.3 g/kg dibasic calcium phosphate, 8.2 g/kg calcium carbonate, and 0.4 g/kg magnesium oxide. KD at 0.3 calories/200 µl was prepared by diluting 11 g of KD in 50 ml of $H_2O$ and stirring for 1 min at about 95°C.

### Confocal microscopy of brains

Brains were dissected under a light microscope in a drop of fresh, ice cold 4% formaldehyde on a Sylgard plate with #5 Dumont forceps (Fine Science Tools) and transferred to an Eppendorf tube containing 1 ml 4% formaldehyde on ice for

approximately 30 min. Brains were washed 3 x 20 min in 1.5 ml 1X phosphate buffered saline (PBS) with 0.3% Triton-X (PBS-T), blocked in 500 µl PBS-T and 5% normal goat serum (Sigma) overnight at 4°C. Block was removed and a primary antibody solution was added and incubated overnight at 4°C. Brains were washed 3 x 20 min in 1.5 ml PBS-T at room temperature. Secondary antibody solution was added and incubated overnight at 4°C. Brains were washed 3 x in 1.5 ml PBS-T at room temperature. The first wash (10 min) contained DAPI in PBS-T, the second wash (30 min) contained Bodipy or Nile Red in PBS-T, and the third wash (10 min) was only PBS-T. During washes and incubations, tubes were placed in a light tight box and agitated. Brains were mounted in Vectashield (Vector Laboratories) and imaged at 100x magnification on a Nikon A1R-SI+ confocal microscope (Optical Imaging Core, University of Wisconsin, Madison, WI). Primary antibodies used were α-Repo (1:100; Developmental Studies Hybridoma Bank), α-GFP (1:500; Invitrogen), α-tubulin (12G10 1:100; Developmental Studies Hybridoma Bank). Fluorescently labeled secondary antibodies used were α-mouse Alexa Fluor 488 (1:500; Invitrogen), α-chicken Alexa Fluor 488 (1:500; Invitrogen), α-mouse Alexa Fluor 594 (1:500; Invitrogen). Fluorescent dyes used were DAPI (1 mg/mL in $H_2O$, diluted 1:1000 in PBS-T, Santa Cruz Biotechnology), Bodipy 493/503 (0.5 mg/mL in DMSO, diluted 1:1000 in PBS-T, Invitrogen), and Nile Red 515/560 (1 mg/mL in DMSO, diluted 1:1000 in PBS-T, Sigma).

## Quantitation and statistical analysis of LDs

Nikon Elements Imaging Software (Optical Imaging Core, University of Wisconsin, Madison, WI) was used to determine the area of individual LDs as well as the number of LDs and nuclei (cells) in each image. Cellpose software was used to segment and count the number of DAPI-positive cells per image [89]. 100x single channel confocal images were used for this analysis from at least 11 brains and 11,000 cells. Approximately half of the brains were imaged from the posterior side and the other half from the anterior side. LDs were sorted by area using Microsoft Excel, and those between 0.2 and 15.0 µm$^2$ were counted. GraphPad Prism (version 8.4.3) was used for graphing and statistical analyses of data in Fig 3 and Tables 1 and 2.

## Lipidomic analysis

Male $w^{1118}$ flies were collected at 0–1 days, aged 7 days and were subject to TBI via the method described in Katzenberger et al., 2013 [47]. Flies were transferred to vials with CMYD for 1, 7, 14, or 21 days. Heads were isolated by freezing flies in liquid nitrogen, vortexing to cause decapitation, and separating heads from bodies by passing them through a sieve. Heads from injured and uninjured flies were stored at −80°C until all samples were ready for analysis. For each time point, three samples of 150 uninjured and three samples of 150 injured heads were submitted for lipidomic analysis (Mass Spectrometry Facility, University of Wisconsin, Madison, WI).

Lipid extraction was performed using a method based extensively on Matyash et al. 2008 [90]. A solution containing 250 µl PBS, 225 µl methanol containing internal standards and 750 µl methyl tert-butyl ether (MTBE) was used for the extraction. The internal standard mix contained Avanti SPLASH LipidoMix (cat #330707−1EA) at 10 µl per sample, supplemented with C18 ceramide-d7 (d18:1-d7/18:0, Cayman Chemical cat #22788), EOS-d9 (d18:1-d9/32:0/18:2, Cayman Chemical cat #24423), linoleic acid-d11 (Cayman chemical cat #9002193), and heptadecanoyl-L-carnitine-d3 (Cayman Chemical cat #35459). Samples of 150 fly heads, held on dry ice, were transferred to bead-beating tubes. To each sample was added 250 µl PBS, 10 µl of internal standard mix, and 215 µl of methanol. The methanol containing internal standards was made up in sufficient volume for all the samples. Samples were then subjected to 2 cycles of homogenization (TissueLyzer II, Qiagen) at 30 Hz for 40 s, rested for 5 min at 4°C, then subjected to 2 more cycles of homogenization. After homogenization, 750 µl MTBE was added and two cycles of extraction, again at 30 Hz for 40 s each with 5 min at 4°C in between, were performed. Following extraction there was a final rest at 4°C for 15 min. Samples were then centrifuged at 17,000 X g for 5 min at 4°C. 700 µl of the upper MTBE phase was transferred to a new 1.5 ml tube and evaporated to

dryness using a speed-vac concentrator. Dried lipid samples were then reconstituted in 150 µl isopropyl alcohol (IPA). After reconstitution in IPA samples were again centrifuged at 17,000 X g for 15 min at 4°C. Concurrently with sample extractions, a process blank and a process blank spiked with internal standard mix were prepared. During data collection, an aliquot of the NIST standard reference material SRM-1950 (http://www-s.nist.gov/srmors/view_detail.cfm?srm=1950), Metabolites in Plasma, was analyzed along with samples and process blanks to evaluate instrument performance. All reconstituted extracts were stored at −80°C prior to analysis. LC/MS/MS analysis of the lipid samples was carried out as described by Landowski *et al.* 2023 [91].

Lipidomic data were analyzed using MetaboAnalyst (metaboanalyst.ca). Raw data for positive and negative ion modes were transferred separately to Microsoft Excel. Only the columns containing the compound names and lipidomic data for the samples were retained; all others were removed. The data were sorted alphabetically by compound name. Group labels were added for each sample. All internal standard species with (d7) and (d9) labels were removed. Duplicate samples were removed, and unique identifiers (*e.g.*, _1 and _2) were added for lipid species with the same name but different peak intensities. Peak intensities were normalized to internal standards by averaging peak intensities for each lipid species (*i.e.*, each row was averaged) and dividing each sample by the average. Normalized values for samples were then averaged (*i.e.*, each column was averaged) and each lipid species was divided by the average. Data from positive and negative ion modes were combined into a single file, and unique identifiers were added for lipid species with the same name. Normalized data were saved as a CSV file and analyzed by Partial Least Square-Discriminant Analysis (PLS-DA) using MetaboAnalyst 6.0, which generated Figs 5A–5D. Data for 1 day (Figs 5A and 5B) were analyzed separately from data for 7, 14, and 21 days (Figs 5C and 5D). S2 and S3 Data were also used for the analyses shown in S2 Fig. Graph-Pad Prism (version 8.4.3) was used to analyze and graph the data in S2 Fig.

## Supporting information

**S1 Fig. Bodipy and Nile Red colocalize in the brain.** Uninjured (left column) and injured (right column) 8-day old, $w^{1118}$ flies were stained with Bodipy (top row, green) and Nile Red (middle row, white) in succession, 24 h post-injury. Yellow in the bottom row (merge) indicates overlap of Bodipy and Nile Red signals, and blue indicates DNA (DAPI).
(TIF)

**S2 Fig. The abundance of certain lipid families differs between the heads of injured and uninjured flies over time after TBI.** Heat maps of lipid class levels over time after TBI. Lipidomic analysis of heads was performed at 1, 7, 14, and 21 days post-injury of 7-day old, male, $w^{1118}$ flies. **(A)** Each box represents the $\log_2$-tranformed ratio of the total abundance of all lipid species within a lipid class in injured versus uninjured flies. **(B)** and **(C)** Each box shows the $\log_2$-transformed ratio of lipid class abundance at 7, 14, and 21 days post-injury, normalized to 1 day levels in **(B)** uninjured and **(C)** injured flies. Dark blue is upregulated, and white is downregulated. Abbreviations: acylcarnitine (CAR), ceramide (Cer), cardiolipin (CL), diacylglycerol (DG), ether-linked phosphatidylcholine (Ether PC), ether-linked phosphatidylethanolamine (Ether PE), fatty acid (FA), fatty acid esters of hydroxy fatty acids (FAHFA), hydroxybutyl monophosphate phospholipid (HBMP), dihexosylceramine (Hex2Cer), lysophosphatidylglycerol (LPC), lysophosphatidylethanolamine (LPE), lysophosphatidylglycerol (LPG), lysophosphatidylinositol (LPI), N-acylethanolamine (NAE), phosphatidylcholine (PC), phosphatidylethanolamine (PE), phosphatidylglycerol (PG), phosphatidylinositol (PI), phosphatidylserine (PS), sterol ester (SE), sphingomyelin (SM), and sterol (ST).
(JPG)

**S1 Data. Lipid droplet size, number per cell, and area per cell over time after TBI and in response to different diets after TBI.** Columns indicate the experiment condition, and rows indicate average values from single microcopy images of many cells in a region of the fly brain. Up to four regions were analyzed for each brain, as described in Fig 1.

Values in Tables 1A and 2A are µm$^2$, in Tables 1B and 2B are number of LDs per cell, and in Tables 1C and 2C are µm$^2$ per cell. These data were used to determine averages and SEMs in Fig 3 and Tables 1 and 2.
(XLSX)

**S2 Data. Lipid species detected in fly heads at 1 day post-injury.** Columns represent the three replicates for both uninjured and injured flies, while rows show the abundance of individual lipid species, shorted alphabetically. These data were used for the analyses shown in Figs 5A and 5B.
(XLSX)

**S3 Data. Lipid species detected in fly heads at 7, 14, and 21 days post-injury.** Each column represents one of three replicates from uninjured or injured flies, and each row corresponds to the abundance of individual lipid species, sorted alphabetically. These data were used for the analyses shown in Figs 5C and 5D.
(XLSX)

## Acknowledgments

We thank Kurt Weiss or assistance with confocal microscopy and image quantitation, Michael Landowski and Aki Ikeda for assistance analyzing the lipidomic data, and Becky Katzenberger for assistance generating S2 Fig. We also thank members of the Ganetzky joint lab meeting for technical and intellectual insights that greatly improved this work.

## Author contributions

**Conceptualization:** Stacey A Rimkus, Barry Ganetzky, David A. Wassarman.

**Data curation:** Stacey A Rimkus.

**Formal analysis:** Stacey A Rimkus, David A. Wassarman.

**Funding acquisition:** Barry Ganetzky, David A. Wassarman.

**Investigation:** Stacey A Rimkus.

**Methodology:** Stacey A Rimkus.

**Project administration:** David A. Wassarman.

**Supervision:** David A. Wassarman.

**Validation:** Stacey A Rimkus.

**Visualization:** Stacey A Rimkus.

**Writing – original draft:** David A. Wassarman.

**Writing – review & editing:** Stacey A Rimkus, Barry Ganetzky, David A. Wassarman.

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
