## [Decision Letter · Decision Letter 0]

18 Jul 2025

PONE-D-25-28751Traumatic brain injury reprograms lipid droplet metabolism shaped by aging and diet in Drosophila brainPLOS ONE

Dear Dr. Wassarman,

Thank you for submitting your manuscript to PLOS ONE. After careful consideration, we feel that it has merit but does not fully meet PLOS ONE’s publication criteria as it currently stands. Therefore, we invite you to submit a revised version of the manuscript that addresses the points raised during the review process.

We look forward to receiving your revised manuscript.

Kind regards,

Firas H Kobeissy, PhD

Academic Editor

PLOS ONE

Journal Requirements:

2. Thank you for stating the following financial disclosure: [Research reported in this publication was supported by the National Institute of Neurological Disorders and Stroke of the National Institutes of Health under Award Number RF1NS114359 and by the UW-Madison School of Medicine and Public Health, Graduate School, and Department of Medical Genetics.]. 

4. We notice that your supplementary files are uploaded with the file type 'Figure'. Please amend the file type to 'Supporting Information'. Please ensure that each Supporting Information file has a legend listed in the manuscript after the references list.

5. We notice that your supplementary files are included in the manuscript file. Please remove them and upload them with the file type 'Supporting Information'. Please ensure that each Supporting Information file has a legend listed in the manuscript after the references list.

Reviewers' comments:

Reviewer's Responses to Questions

**Comments to the Author**

1. Is the manuscript technically sound, and do the data support the conclusions?

Reviewer #1: Yes

Reviewer #2: Partly

2. Has the statistical analysis been performed appropriately and rigorously? 

Reviewer #1: Yes

Reviewer #2: Yes

3. Have the authors made all data underlying the findings in their manuscript fully available?

Reviewer #1: Yes

Reviewer #2: Yes

4. Is the manuscript presented in an intelligible fashion and written in standard English?

Reviewer #1: Yes

Reviewer #2: Yes

5. Review Comments to the Author

Reviewer #1: The manuscript by Rimkus et al. reports a detailed investigation of brain lipid metabolism dynamics in a drosophila model of TBI. Previous investigations from this group are consistent with results obtained from other groups that investigate the fruit fly to model TBI using different injury mechanisms. The application of lipidomics is novel to fly TBI models and complements the advanced imaging techniques. The data from all fly-TBI groups is, wherever comparisons are possible, also consistent with the comparatively low-resolution knowledge obtained from mammalian models.

This manuscript is well written. The authors go to great lengths in describing their results in detail which may render the manuscript somewhat cumbersome in some sections.

Comments

Abstract line 34-37 how do we know that TBI selectively impairs LD production as opposed to LD degradation?

Introduction: line 58-59. This is an important statement and this reviewer would have appreciated higher-level references to substantiate it. A similar concern applies to reference 11, published in Heliyon. These references to somewhat nebulous review articles are in contrast to the high-quality references in the subsequent paragraphs.

What is the role of proteins in the formation and maintenance of LDs and could impaired protein (e.g. apolipoproteins) synthesis result in the observed changes?

Therefore, the described changes in lipid metabolism described in this manuscript are a valuable addition to the fund of knowledge about mechanism of the secondary injury after TBI.

Side note: ‘cargo€s is inconsistently spelled in the discussion.

Line 365 ff: the references to clinical nutrition guidelines may be correct but the causality – use of saline instead of glucose-containing solutions is inaccurate. The reasoning was primarily avoidance of hypoosmotic solutions. It may be a bit too far stretched to cite antiquated clinical nutrition guidelines (which reflect decades old and frequently obsolete clinical insights) in a cutting-edge research paper. Supporting it with a ‘narrative review’ (the least rigorous form of review ref 63) only weakens the argument. I would suggest leaving ref 62 and removing the rest of the paragraph lines 371-374.

Reviewer #2: The manuscript by Rimkus et al describes changes in lipids and lipid droplets that occur after TBI, suggesting that these could contribute to the susceptibility to develop neurodegeneration later. A major issue is that although they detect a significant increase in LD number 21d after the injury, these are not detectable after further aging. Furthermore, the area of LDs is not significantly different at any of the later timepoints. Concerning the lipids, although there are also differences at 21d, later timepoints have not been analyzed and so it is unknown whether changes in the lipid composition persist that could be the basis for the increased susceptibility. Another concern is that the statement that there is abnormal trafficking of microtubule is not supported by experimental data and it is difficult to judge the effects on microtubule in figure 4A and B; although at 14d the number of LDs should be about the same in uninjured and injured flies from their data, there are more LDs in the picture of the uninjured flies and it is not clear whether the pictures are from comparable regions of the brain. The authors point out that it is an important question whether the LDs are in glia or neurons which could be easily addressed by neuronal markers.

6. PLOS authors have the option to publish the peer review history of their article (what does this mean?). If published, this will include your full peer review and any attached files.

Reviewer #1: No

Reviewer #2: No

---

## [Author Response · Author response to Decision Letter 1]

25 Jul 2025

Reviewer #1: The manuscript by Rimkus et al. reports a detailed investigation of brain lipid metabolism dynamics in a drosophila model of TBI. Previous investigations from this group are consistent with results obtained from other groups that investigate the fruit fly to model TBI using different injury mechanisms. The application of lipidomics is novel to fly TBI models and complements the advanced imaging techniques. The data from all fly-TBI groups is, wherever comparisons are possible, also consistent with the comparatively low-resolution knowledge obtained from mammalian models. This manuscript is well written. The authors go to great lengths in describing their results in detail which may render the manuscript somewhat cumbersome in some sections.

Comments

Abstract line 34-37 how do we know that TBI selectively impairs LD production as opposed to LD degradation?

Although our data do not definitively distinguish whether TBI affects LD production versus degradation at 21 days post-injury, our lipidomic results – along with findings from other studies (references 32, 47, 55, and 75) – support a role for impaired LD production. Therefore, we believe the interpretation that ‘These findings suggest that TBI selectively impairs age-dependent production of new LDs without affecting the growth of existing LDs’ is accurate, while still allowing for the possibility that degradation may also be involved.

Introduction: line 58-59. This is an important statement and this reviewer would have appreciated higher-level references to substantiate it. A similar concern applies to reference 11, published in Heliyon. These references to somewhat nebulous review articles are in contrast to the high-quality references in the subsequent paragraphs.

The reviewer makes a good point. To address this, for TBI, we cited primary research articles and review articles that highlight mitochondrial dysfunction, oxidative stress, and neuroinflammation as early contributors to TBI outcomes (references 9-18). For neurodegenerative disorders, we cited reviews discussing roles of these same mechanisms in Alzheimer’s, Parkinson’s, and Huntington’s diseases (references 19-27).

What is the role of proteins in the formation and maintenance of LDs and could impaired protein (e.g. apolipoproteins) synthesis result in the observed changes?

Proteins, including apolipoproteins, are critical for formation and maintenance of LDs and contribute to TBI-induced changes. As noted in lines 116-121, our GWAS data implicate several LD-related proteins in early mortality after TBI; however, the current data are insufficient to propose specific protein-level mechanisms.

Therefore, the described changes in lipid metabolism described in this manuscript are a valuable addition to the fund of knowledge about mechanism of the secondary injury after TBI.

Side note: ‘cargo€s is inconsistently spelled in the discussion.

Cargos was changed to cargoes on page 14.

Line 365 ff: the references to clinical nutrition guidelines may be correct but the causality – use of saline instead of glucose-containing solutions is inaccurate. The reasoning was primarily avoidance of hypoosmotic solutions. It may be a bit too far stretched to cite antiquated clinical nutrition guidelines (which reflect decades old and frequently obsolete clinical insights) in a cutting-edge research paper. Supporting it with a ‘narrative review’ (the least rigorous form of review ref 63) only weakens the argument. I would suggest leaving ref 62 and removing the rest of the paragraph lines 371-374.

We appreciate the insights of the reviewer and have followed their suggestion of deleting lines 371-374 and the accompanying citations.

Reviewer #2: The manuscript by Rimkus et al describes changes in lipids and lipid droplets that occur after TBI, suggesting that these could contribute to the susceptibility to develop neurodegeneration later.

A major issue is that although they detect a significant increase in LD number 21d after the injury, these are not detectable after further aging. Furthermore, the area of LDs is not significantly different at any of the later timepoints.

At 21 days, LD number increased in uninjured but not injured flies. As noted in lines 381-397, this may reflect other phenotypes that emerge around this time in uninjured flies (references 32, 47, 55, and 75). A significant difference in LD area was not only observed at 21 days but also 35 days (Figure 3C).

Concerning the lipids, although there are also differences at 21d, later timepoints have not been analyzed and so it is unknown whether changes in the lipid composition persist that could be the basis for the increased susceptibility.

We agree that additional time points would have helped determined whether changes in lipid composition persist. However, due to the high cost of lipidomic analysis, we were limited in the number of time points that could be included.

Another concern is that the statement that there is abnormal trafficking of microtubule is not supported by experimental data and it is difficult to judge the effects on microtubule in figure 4A and B; although at 14d the number of LDs should be about the same in uninjured and injured flies from their data, there are more LDs in the picture of the uninjured flies and it is not clear whether the pictures are from comparable regions of the brain.

We agree that the images in Figures 4A and B do not clearly support the text due to differences in microtubule orientation. To address this, we replaced Figure 4B with one that matches the orientation of Figure 4A.

The authors point out that it is an important question whether the LDs are in glia or neurons which could be easily addressed by neuronal markers.

It is possible to determine whether LDs are located in glia or neurons by labeling neuronal or glial cell membranes and performing 3D reconstruction. We plan to pursue this in future studies, but it is beyond the scope of the current manuscript.

---

## [Decision Letter · Decision Letter 1]

29 Aug 2025

Traumatic brain injury reprograms lipid droplet metabolism shaped by aging and diet in Drosophila brain

PONE-D-25-28751R1

Dear Dr. Wassarman,

We’re pleased to inform you that your manuscript has been judged scientifically suitable for publication and will be formally accepted for publication once it meets all outstanding technical requirements.

Kind regards,

Firas H Kobeissy, PhD

Academic Editor

PLOS ONE

Additional Editor Comments (optional):

Reviewer #1:

Reviewer #2:

Reviewers' comments:

Reviewer's Responses to Questions

**Comments to the Author**

1. If the authors have adequately addressed your comments raised in a previous round of review and you feel that this manuscript is now acceptable for publication, you may indicate that here to bypass the “Comments to the Author” section, enter your conflict of interest statement in the “Confidential to Editor” section, and submit your "Accept" recommendation.

Reviewer #1: All comments have been addressed

Reviewer #2: All comments have been addressed

2. Is the manuscript technically sound, and do the data support the conclusions?

Reviewer #1: Yes

Reviewer #2: Yes

3. Has the statistical analysis been performed appropriately and rigorously? 

Reviewer #1: Yes

Reviewer #2: Yes

4. Have the authors made all data underlying the findings in their manuscript fully available?

Reviewer #1: Yes

Reviewer #2: Yes

5. Is the manuscript presented in an intelligible fashion and written in standard English?

Reviewer #1: Yes

Reviewer #2: Yes

6. Review Comments to the Author

Reviewer #1: I still consider the Heliyon reference as 'weak' and think that clinical references do not always strengthen lab bench work in IMOs.

Reviewer #2: The authors have added another figure to address my concerns that the previous picture did not clearly show what they described. The other critiques have also been addressed sufficiently to justify publication.

7. PLOS authors have the option to publish the peer review history of their article (what does this mean?). If published, this will include your full peer review and any attached files.

Reviewer #1: No

Reviewer #2: No

---

## [Editor Report · Acceptance letter]

PONE-D-25-28751R1

PLOS ONE

Dear Dr. Wassarman,

I'm pleased to inform you that your manuscript has been deemed suitable for publication in PLOS ONE. Congratulations! Your manuscript is now being handed over to our production team.

Kind regards,

on behalf of

Dr. Firas H Kobeissy

Academic Editor

PLOS ONE